# Prevalence of stunting and its associated factors among children 6–59 months of age in pastoralist community, Northeast Ethiopia: A community-based cross-sectional study

Mulugeta Gebreayohanes[1], Awrajaw Dessie[2]*

**1** Department of Human Nutrition, Institute of Public Health, University of Gondar, Gondar, Ethiopia,
**2** Department of Environmental and Occupational Health and Safety, Institute of Public Health, University of Gondar, Gondar, Ethiopia

* awrajawdss@gmail.com

**Data Availability Statement:** All relevant data are within the paper and its Supporting information files.

## Abstract

### Introduction

Globally, stunting is a significant public health concern and it is very critical in Ethiopia. This research aims to determine the prevalence of stunting and its correlates among children in the pastoral community.

### Methods

A community-based cross-sectional study was conducted in Dubti District, Afar Region, North East Ethiopia from 2–31 January 2018. A total of 554 children were recruited using a multi-stage sampling technique and participated in this study. A binary logistic regression analysis was performed to determine factors linked to stunting. The significance of the associations was determined at a p-value < 0.05 and the adjusted odds ratio at 95% CI was calculated to evaluate the strength of the associations.

### Results

The prevalence of stunting was 39.5% (95% CI: 35.4–43.5%). The odds of stunting were increased, so does the age of the child increased as compared to 6–11 months of children. Initiating breastfeeding after 1 hour after birth (AOR = 1.99; 95% CI: 1.22, 3.23), not exclusively breastfeeding for at least 6 months (AOR = 2.57; 95% CI: 1.49, 4.42), poor dietary diversity (AOR = 1.93; 95% CI: 1.03, 3.62), and using unprotected water for drinking (AOR = 1.68; 95% CI: 1.21, 2.94) were significant factors.

### Conclusion

Among children aged 6–59 months, the level of stunting in the pastoral community was significantly high. The study found that stunting was associated with multiple nutritional and non-nutritional factors. To tackle stunting, inter-sectoral cooperation is needed by enhancing

**Funding:** The authors received no specific funding for this work.

**Competing interests:** The authors have declared that no competing exist.

**Abbreviations:** ANC, Antenatal Care; AOR, Adjusted Odds Ratio; CI, Confidence Interval; COR, Crude Odds Ratio; GDP, Gross Domestic Product; GTP, Growth Transformation Plan; HAZ, Height for Age Z-score; HSTP, Health Sector Transformation Plan; IQR, Inter Quartile Range; MDDS, Minimum Dietary Diversity Score; NGO, Non-Governmental Organization; PNC, Postnatal Care; SD, Standard Deviation; SDG, Sustainable Development Goal; SPSS, Statistical Package for Social Science; TLU, Tropical Livestock Unit; WHO, World Health Organization.

the clean water supply of the community, optimal breastfeeding practice, food diversity, and economic status.

# Background

One of the most important causes of child morbidity and mortality in developing nations is malnutrition. Sub-Saharan Africa leads by high child morbidity and death rates associated with malnutrition [1]. It affects a nation's future economic competitiveness and continues to be a key concern in developing countries [2]. Stunting is chronic malnutrition that occurs when children struggle to meet their capacity for linear growth and cause significant physical and cognitive damage [3]. It will also have irreparable effects on a child's future growth, which will increase population vulnerability and weaken its capacity to cope with episodes of food stress [4].

Over 161 million children under the age of five are affected by stunting worldwide, with an estimated one million deaths. African and Asian children are hit hard by stunting [3]. Under-nutrition is one of the major health issues of children under the age of five and results in a 16.5% GDP loss in Ethiopia [5, 6]. Stunting can have devastating health and economic costs that last for a lifetime [7].

With very weak human development metrics, Ethiopia is one of the poorest countries in the world. Around 23 million Ethiopians live under the poverty line and food insecurity remains a major problem [8]. Pastoralists (90%) dominate the production system of the Afar region, from which agro-pastoralists (10%) now emerge after some permanent and temporary rivers on which small-scale irrigation grows. Strong reminiscences of suffering are due to the frequent episodes of drought and unseasoned flooding and disease outbreaks in the pastoral areas of Ethiopia. The reduction of hunger, food security, and pastoral livelihood strategies are therefore largely dependent on the climate system and vulnerable to seasonal variations.

The prevalence of stunting in Ethiopia has dropped considerably from 58% in 2000 to 38% in 2016, but in the Afar region, the stunting level is above the national average of 41% [9]. Malnutrition in the Afar region poses a huge problem [10]. While the consequences of stunting are clear, its causes are more complex [11]. Poor nutritional and health condition of a mother, insufficient infant and young child feeding practices, micronutrient deficiencies, and infection are primary factors leading to stunting [12].

The United nation' sustainable development goals (SDGs) have marked stunting along with other nutrition indicators as the main focus areas to eradicate global malnutrition [13]. Stunting is regarded by the Ethiopian government as a major public health issue and an obstacle to its economic goals. The Health Sector Transformation Plan (HSTP), part of GTP II, aimed at reducing mortality rates of 30 per 1,000 live births below five years in Ethiopia, reducing stunting to 26% in less than 5 years [14].

To tackle stunning on a sustainable basis, it is important to understand local geo-cultural domains such as tradition and community livelihoods. In this context, it is also important to interpret prevalence and cause factors. Access to seasonal pastures is prioritized by these communities and they are highly mobile between different wet and dry seasons to seek food for their livestock, their main livelihoods, in a much-dispersed way. In pastoralist groups, however, there is a lack of evidence regarding the extent of stunting and its correlates. Therefore, this research in the Dubti district could reflect the effectiveness of a permanent solution for addressing stunting in the region by pointing out the main determinants of stunting, rather than relying on targeting short-term food help that would ultimately not overcome stunting. This study will therefore provide input to local government officials, non-governmental

organizations (NGOs), policymakers working to reduce the rate of child mortality that contributes to the goal of the Health Sector Transformation Plan (HSTP) and Growth Transformation Plan (GTP) II of Ethiopia on the prevalence and related factors for children aged 6–59 months.

## Materials and methods

### Study design and period

A community-based cross-sectional study was conducted from 2–31 January 2018.

### Study area

The study was conducted in the Dubti district, Afar region, Northeast Ethiopia. It is one of the 32 districts in the Afar region. It is located approximately 7 km far from the regional capital Semera [15] and 600 km northeast of Addis Ababa, the capital of Ethiopia [16]. The study region is often struck by drought. The community is predominantly pastoralists with a small plot of land for cultivation and is primarily engaged in the rearing or husbandry of livestock. They live in a scattered way and many places remain isolated and difficult to enter. Pastorals are particularly susceptible to extensive droughts. Infrastructure or facilities such as water, sanitation, basic health, and nutrition are very limited in terms of accessibility [15]. According to the district health center annual report, a total of 8187 under-five children resided in the area [17].

### Source and study population

Children aged 6–59 months living in the district were the source populations of this study and the study population consisted of children from 4 (3 rural and 1 urban) randomly selected Kebeles in the district. During the entire data collection season, children who were critically ill and those affected by spinal curvature (kyphosis, scoliosis, and kyphoscoliosis) were exempted from the study.

### Sample size determination and sampling technique

The sample size was determined by a single population proportion formula using the assumptions of 95% confidence level, the proportion of stunting among 6–59 months children in Afambo district 32.2% [18], 5% margin of error, and a design effect of 1.5. The final sample size was therefore calculated to be 555, after a 10% non-response rate was added. The study participants were recruited using a multi-stage sampling technique. In the first stage, from the two urban kebeles, one kebele was selected by the lottery method, and in the same way, from the 12 rural kebeles, three kebeles were selected. Second, the total sample size was allocated proportionally based on the total number of households with children aged 6–59 months and a simple random sampling technique was used to select children based on the existing sampling frame from health posts. The index child or the youngest child was selected in this study from households with two or more children aged 6–59 months. The mother or guardians were interviewed.

### Variables measurement

**Height and weight.** The height of infants aged between 6 months and 23 months was measured in a recumbent position to the nearest 0.1 cm using a board with an upright wooden base and a movable headpiece. Children between 24 and 59 months of age were measured in a standing position of 0.1 cm to the nearest. Besides, the child weight was measured using an

electronic digital weight scale for children who were comfortable to measure on their own, and also for children who were uncomfortable to measure on their own, we used the combined mother and child weight and the mother's weight to calculate the child's weight [19].

**Stunting.** Height-for-age is a measure of linear growth retardation and cumulative growth deficits. Children whose height-for-age Z (HAZ) score is below minus two standard deviations (-2 SD) from the median of the reference population were considered to be stunted. Children below minus three standard deviations (-3 SD) were considered to have been severely stunted [9].

**The economic status of households.** Since the community is pastoral, the economic status of households has been measured using the Tropical Livestock Unit (TLU) as a proxy. TLU is a measure developed by the Food and Agriculture Organization (FAO) that allows the combination of multiple animal species into a weighted measure of total body weight and potential market value [20]. A single animal weighing 250 kg is a single TLU, which provides a weighting factor of 0.7 for cattle, 0.1 for sheep, 0.1 for goats, and 0.01 for chickens. The economic status of households was determined by comparing the TLU scores to the standard ranking. A score below 5 TLU shows the household is poor. A TLU score of 5 to 12.99 showed the household's economic status was medium and richer households ranked 13 and above TLU [20].

**Minimum Dietary Diversity Score (MDDS).** Proxy for the adequacy of dietary micronutrient density for infants and young children. Consumption of 4 or more of the 7 food groups means that the child is likely to consume at least one animal food source and at least one fruit or vegetable in addition to the staple food (grains, roots, or tubers) in the last 24 hours. Four food groups should be drawn from the list of seven food groups: grains, roots and tubers, legumes and nuts; dairy products (milk yogurt, cheese); meat, fish, poultry, and liver / organic meat; eggs; vitamin A-rich fruits and vegetables; and other fruits and vegetables [18].

**Fully vaccinated**: Children who had received a vaccination against tuberculosis (BCG), three doses each of the DPT and polio vaccines, and a measles vaccination by the age of 12–23 months [21]. **Non-vaccinated**: Children who had not received a vaccination against tuberculosis (BCG), three doses each of the DPT and polio vaccines, and a measles vaccination by the age of 12–23 months. **Partially vaccinated**: Children who had started the vaccination but not completed all the doses due to forgetfulness, and drop out [21].

## Data quality management

The structured questionnaire was prepared in English and translated into the Afar language and translated back into English by language experts to check its consistency. The pre-test of the questionnaire was performed on 5% of the sample size in a similar area-Asayita district, Afar region. The weight measurement scale was checked against zero readings after and before each child was weighed. A two-day training was given to data collectors and supervisors on processes, techniques, and methods for collecting data. In addition, a clear introduction was given to respondents on the intent and objectives of the study before data collection. In parallel, constant and strict monitoring and on-the-spot checks have been carried out throughout the process.

## Data processing and analysis

The data were verified, coded, and entered in version 7.2 of Epi-Info software. Sex, age, and weight data were transferred to the WHO Anthro software using the WHO standard with participant identification numbers to translate the nutritional data into Z scores for the HAZ indices. The data, including the HAZ, was subsequently exported to SPSS version 20 for analysis. The bivariate analysis was performed to determine the association of stunting and associated

factors and the variables were selected for multivariate analysis by p-value < 0.2. A multivariable binary logistic regression analysis was employed to control the possible effects of confounders. The model goodness of fit test was checked by Hosmer and Lemeshow Test, and it was found fit ($X^2$ = 11.57, p-value = 0.17). Finally, variables that showed significant associations were identified based on the adjusted odds ratios (AOR) with a 95% CI and p-value <0.05.

### Ethical statement

This study was approved by the Institutional Review Board of the University of Gondar, Institute of Public Health, Ethiopia. Informed consent was obtained from all parents/ or legal guardian of children participated in this study after adequate explanations of the study. For illiterate parents/ or legal guardians and informed consent was obtained from their legally Authorized Representatives. The study was conducted according to the Declaration of Helsinki, and the National Research Ethics Review Guideline for Medical and Health Research Involving Human Subjects, enforced by the Ministry of Science and Technology, Government of Ethiopia.

## Results

### Demographic and socio-economic characteristics

The study included a total of 554 study participants, giving a response rate of 99.82%. The majority of households were male-headed (92.1%). Approximately 92.6% of respondents were Muslim, and more than three-fourth (79.1%) of the participants were Afar by ethnicity. Five hundred and seven mothers (91.5%) were married and 79.2% were aged between 20–34 years. More than half of respondents (57.9%) can't read and write. Nearly three-fourths of households have been classified as poor based on TLU. The average household family size was 5 (SD± 1.81), and 45.3% of households had fewer than 5 family members. Nearly half (48%) and 46.4% of households had 1 and 2 children under five years of age, respectively. Most of the mothers were housewives (62.5%), and 81.6% of the fathers were agro-pastoralists by occupation. The majority of households (92.8%) were not productive safety net program (PSNP) users (Table 1).

### Child characteristics and child carrying practice

This study showed that 56.3% of the children were males and 20.0% of them were aged 6–11 months with a median (IQR) age of 31 (IQR = 19) months. More than one-third of children were second by birth order and 99.1% of the children were singletons. More than half (52.5%) of the children were born at home, according to this study. The study revealed that 79.2% of mothers started breastfeeding with colostrum within one hour immediately after birth and the majority (80.5%) of the children were breastfed exclusively for at least six months. This study found that 54.7% children had of at least 3 meal frequencies a day and the majority (83.2%) of children had a minimum dietary diversity score of < 4. About 59.8% of children have been fully vaccinated. Regarding the morbidity status of the children, 65.7% of the children had at least one disease. More than half (59.9%) of the children encountered acute respiratory infection in the past two weeks before the data collection. Moreover, 14.4% and 13% of the children were also affected by diarrhea and stomach illness, respectively. Malaria and measles were also reported in 3.2% and 4.0% of the children in this study (Table 2).

### Maternal characteristics and health service utilization

This study shows that more than half of mothers were aged between 26 and 35 years when they gave birth to the index child, with a median (IQR) age of 28 (IQR = 6) years. About 50.2% completed the full ANC schedule and 75.1% received PNC. More than one-third of mothers

**Table 1. Demographic and socio-economic characteristics of households in Dubti district, Afar region, northeast Ethiopia, January 2018 (N = 554).**

| Variables | Category | Frequency | Percent |
|---|---|---|---|
| Head of Household | Father of the child | 510 | 92.1 |
| | Mother of the child | 25 | 4.5 |
| | Others* | 19 | 3.4 |
| Ethnicity | Afar | 438 | 79.1 |
| | Amhara | 86 | 15.5 |
| | Oromo | 15 | 2.7 |
| | Others# | 15 | 2.7 |
| Religion | Muslim | 513 | 92.6 |
| | Orthodox | 24 | 4.3 |
| | Protestant | 17 | 3.1 |
| Marital status | Married | 507 | 91.5 |
| | Unmarried | 13 | 2.3 |
| | Divorced/Widowed/separated | 34 | 6.1 |
| Total number of <5 children | 1 | 266 | 48.0 |
| | 2 | 257 | 46.4 |
| | 3 | 31 | 5.6 |
| Educational status of mother | Can't read and write | 321 | 57.9 |
| | Informal education | 21 | 3.8 |
| | Primary education | 129 | 23.3 |
| | Secondary education | 56 | 10.1 |
| | Higher education | 27 | 4.9 |
| Occupational status of mother | Housewife | 347 | 62.6 |
| | Agro-pastoralist | 150 | 27.1 |
| | Merchant | 34 | 6.1 |
| | Others§ | 23 | 4.2 |
| Occupational status of father | Agro-pastoralist | 452 | 81.6 |
| | Merchant | 68 | 12.3 |
| | Government/self-employee | 34 | 6.1 |
| Family size | <5 | 251 | 45.3 |
| | ≥5 | 303 | 54.7 |
| Wealth status | Poor | 412 | 74.4 |
| | Medium | 81 | 14.6 |
| | Rich | 61 | 11.0 |
| PSNP user | No | 514 | 92.8 |
| | Yes | 40 | 7.2 |

Note: Others:

*Caregivers of the targeted child;

# Tigre, Wolayita, Somali;

§ Private organization employee, government employee, student, NGO employee. PSNP = Productive safety net programme

(65.9%) had no extra meal at all during their pregnancy or lactation. Husbands make decisions in the majority of households concerning the use of money (71.3%) (Table 3).

## Environmental health characteristics of households

The majority (62.3%) of households used a public tap as a source of drinking water, which is one of the improved sources of drinking water. About 48.2% of households had access to water

**Table 2. Child characteristics and child carrying practice in Dubti district, Afar region, northeast Ethiopia, January 2018 (N = 554).**

| Variables | Category | Frequency | Percent |
|---|---|---|---|
| Sex of the child | Female | 242 | 43.7 |
| | Male | 312 | 56.3 |
| Age of the child in months | 6–11 | 111 | 20.0 |
| | 12–23 | 122 | 22.0 |
| | 24–35 | 177 | 31.9 |
| | 36–59 | 144 | 26.0 |
| Birth order | First | 122 | 22.0 |
| | Second | 198 | 35.7 |
| | Third | 110 | 19.9 |
| | Fourth and above | 124 | 22.4 |
| Type of birth | Single | 549 | 99.1 |
| | Twin | 5 | 0.9 |
| Place of delivery | Home | 291 | 52.5 |
| | Health institution | 263 | 47.5 |
| Time of initiation of breastfeeding | Within 1 hour | 439 | 79.2 |
| | After 1 hour | 115 | 20.8 |
| Exclusive breastfeeding | <6 months | 108 | 19.5 |
| | ≥6 months | 446 | 80.5 |
| Frequency of feeding | <3 times | 251 | 45.3 |
| | ≥3 times | 303 | 54.7 |
| Minimum dietary diversity score | <4 | 461 | 83.2 |
| | ≥4 | 93 | 16.8 |
| Immunization status (n = 122) | Not vaccinated | 7 | 5.7 |
| | Partially vaccinated | 42 | 34.4 |
| | Fully vaccinated | 73 | 59.8 |
| Morbidity | No disease | 190 | 34.3 |
| | One disease | 253 | 45.7 |
| | Two and more diseases | 111 | 20.0 |

NB: Children aged 12–23 months were considered to compute the frequency of immunization status.

**Table 3. Maternal characteristics in Dubti district, Afar region, northeast Ethiopia, January 2018 (N = 554).**

| Variables | Category | Frequency | Percent |
|---|---|---|---|
| Age of mother | ≤25 | 201 | 36.3 |
| | 26–35 | 279 | 50.4 |
| | ≥36 | 74 | 13.3 |
| Number of ANC visit | 0 | 36 | 6.5 |
| | 1–3 | 240 | 43.3 |
| | ≥4 | 278 | 50.2 |
| PNC follow up | No | 138 | 24.9 |
| | Yes | 416 | 75.1 |
| An extra meal is given to the mother during pregnancy or lactation | No | 189 | 34.1 |
| | Yes | 365 | 65.9 |
| Decision making on the use of money | Mainly wife | 81 | 14.6 |
| | Mainly husband | 395 | 71.3 |
| | Both jointly | 78 | 14.1 |

for the round trip within less than 15 minutes. More than half of the participants (59.2%) used the latrine for defecation. From the study, 38.6% of mothers and caregivers washed their hands with water only (Table 4).

## Prevalence of stunting

The prevalence of stunting was found to be 39.5% (95% CI: 35.4–43.5%). Moreover, the prevalence of moderate and severe stunting was 29.6% and 9.9%, respectively. The prevalence of stunting among female and male children was 36.0% and 42.3%, respectively. The highest prevalence of stunting was 61.1% among children aged 36–59 months, 41.2% among those aged 24–35 months, and 34.4% among those aged 12–23 months, and the lowest 14.4% was among children aged 6–11 months. Of stunted children, the majority 41.8% were between the ages of 36–59 months and the minimum 5.5% were between the ages of 6–11 months.

## Factors associated with stunting

Stunting was correlated with economic status, age of the child, breastfeeding initiation, complementary feeding, minimum dietary diversity score, and source of drinking water. Children from poor households were 5.5 times more likely to be stunted than children from a rich families (AOR = 5.50; 95% CI: 2.52, 12.04). Stunting was more common among children aged 12–23, 24–35, and 36–39 months compared to children aged 6–11 months (AOR = 2.55; 95% CI: 1.27, 5.09), (AOR = 3.02; 95% CI: 1.58, 5.78), and (AOR = 4.12; 95% CI: 2.00, 8.45), respectively.

The time breastfeeding initiated after birth and exclusive breastfeeding were among the predictors for stunting in this study. Chances of being stunted have increased by 99% among children who started breastfeeding after 1 hour compared with children who started breastfeeding within 1 hour of birth (AOR = 1.99; 95% CI: 1.22, 3.23). In comparison, infants who exclusively breastfeed for less than 6 months were 2.57 more likely to be stunted than their counterparts who had exclusively breastfed for 6 months or longer (AOR = 2.57; 95% CI: 1.49, 4.42).

Minimum dietary diversity score (MDDS) was found to be linked to stunting. Children from mothers who had 4 and less score were 93% more likely to be stunted than their counterparts who had a score of more than 4 (AOR = 1.93; 95% CI: 1.03, 3.62). Households using unprotected river water were 68% more likely their children to be stunted than households receiving drinking water from public tabs (AOR = 1.68; 95% CI: 1.21, 2.94) (Table 5).

**Table 4. Environmental health characteristics of households in Dubti district, Afar region, northeast Ethiopia, January 2018 (N = 554).**

| Variables | Category | Frequency | Percent |
|---|---|---|---|
| Source of drinking water | River | 150 | 27.1 |
| | Spring | 59 | 10.6 |
| | Public tap | 345 | 62.3 |
| Time to obtain drinking water (round trip) | <15 minutes | 267 | 48.2 |
| | 15–30 minutes | 160 | 28.9 |
| | >30 minutes | 127 | 22.9 |
| Latrine utilization | No | 226 | 40.8 |
| | Yes | 328 | 59.2 |
| Materials used for hand washing | Water only | 214 | 38.6 |
| | Using soap sometimes | 227 | 41.0 |
| | Using soap always | 113 | 20.4 |

**Table 5. Factors affecting stunting among children aged between 6 and 59 months in Dubti district, Afar region, northeast Ethiopia, January 2018 (N = 554).**

| Variables | Stunting | | COR with 95% CI | AOR with 95% CI |
|---|---|---|---|---|
| | Yes | No | | |
| **Economic status** | | | | |
| Poor | 186 | 226 | 4.2 (2.07, 8.49)*** | 5.50 (2.52, 12.00)*** |
| Medium | 23 | 58 | 2.02 (0.88, 4.65) | 2.48 (0.96, 6.15) |
| Rich | 10 | 51 | 1 | 1 |
| **Sex of the child** | | | | |
| Female | 87 | 155 | 0.76 (0.54, 1.08) | 0.68 (0.46, 1.02) |
| Male | 132 | 180 | 1 | 1 |
| **Age of the child** | | | | |
| 6–11 | 16 | 95 | 1 | 1 |
| 12–23 | 42 | 80 | 3.12(1.63,5.96)** | 2.51 (1.25, 5.04)** |
| 24–35 | 73 | 104 | 4.17(2.27,7.66)*** | 2.99 (1.56, 5.73)** |
| 36–59 | 88 | 56 | 9.33(4.99,17.46)*** | 4.11 (2.00, 8.45)*** |
| **Initiation of breastfeeding** | | | | |
| After 1 hour | 66 | 49 | 2.52 (1.66, 3.83)*** | 1.89 (1.17, 3.06)** |
| Within 1 hour | 153 | 286 | 1 | 1 |
| **Exclusive breastfeeding** | | | | |
| ≤6 months | 69 | 39 | 3.49 (2.25, 5.42)*** | 2.51 (1.47, 4.29)** |
| >6 months | 150 | 296 | 1 | 1 |
| **Frequency of feeding per day** | | | | |
| <3 times | 104 | 147 | 1.16 (0.82, 1.63) | 1.03 (0.69, 1.53) |
| 3 times and more | 115 | 188 | 1 | 1 |
| **Minimum dietary diversity score** | | | | |
| ≤4 | 202 | 259 | 3.49 (1.99, 6.09)*** | 1.94 (1.04, 3.64)* |
| >4 | 17 | 76 | 1 | 1 |
| **Time to obtain drinking water (round trip)** | | | | |
| <15 minutes | 88 | 179 | 1 | 1 |
| 15–30 minutes | 67 | 93 | 1.46 (0.98, 2.19) | 1.59 (0.98, 2.58) |
| >30 minutes | 64 | 63 | 2.07 (1.34, 3.18)** | 1.58 (0.83, 3.00) |
| **Materials used for hand washing** | | | | |
| Only water | 103 | 111 | 1.69 (1.06, 2.71)* | 1.33 (0.76, 2.31) |
| Using soap sometimes | 76 | 151 | 0.92 (0.57, 1.48) | 1.04 (0.61, 1.80) |
| Always using soap and water | 40 | 73 | 1 | 1 |
| **Source of drinking water** | | | | |
| River | 82 | 68 | 2.41 (1.63, 3.57)*** | 1.72 (1.01, 2.98)* |
| Spring | 22 | 37 | 1.19 (0.67, 2.11) | 0.88 (0.41, 1.91) |
| Public tap | 115 | 230 | 1 | 1 |

## Discussion

The level of stunting among children 6–59 months children was 39.5% in the current study. Moreover, the prevalence of severe and mild stunting was 9.93% and 29.6%, respectively. Stunting, which is an indicator of chronic malnutrition would result in delayed developmental milestones, inadequate psychosocial stimulation, poor school performance over the years, and a compromised life-course potential [22]. These conditions entirely impacted the progress of all SDG targets. Hence, addressing child nutritional problems is key for national and global health, education, and economic developmental agendas.

The level of stunting found in this study is designated to be very serious or critical in the study area, according to the WHO classification [23], which implied that stunning is the big public health challenge in Ethiopia. The result is in line with the national prevalence of stunting (38%) among under-five children [24]. The prevalence was, however, lower than that of other studies conducted in the Hadibu Abote district, Oromia region, which reported 47.6% prevalence [25]; in the district of Bule Hora, South Ethiopia, 47.6% [26]; and 67.8% in the district of Asayita district, Eastern Ethiopia [27]; 56.6% in the district of Medebay Zana district, Northern Ethiopia [28]; 49.1% in the district of Libo-Kemekem, North-west Ethiopia [29]. However, the finding was higher than the prevalence of stunting which was reported in the Afambo district of Eastern Ethiopia (32.2%) [18]; the study done in Dollo Ado district (34.4%) [30], another study conducted in eastern Ethiopia (34.4%) [31]; a study conducted in Delanta district, Ethiopia (22.1%) [32]. The discrepancies in the finding may be due to differences in the sample size and other socio-economic factors such as feeding habits, policies for infant and child feeding, differences in education and culture. Milk and milk products consumption in the study area could help child development and ultimately tackling stunting, as confirmed in the scientific article [33]. This could be helpful to fight for improving the nutritional status of children in the nomadic community, where a large population of cattle, sheep, goats, and camels are found.

This research has shown that 65.7% of children were affected by at least one disease such as diarrhea, respiratory infections, malaria, and measles, etc This finding has been corroborated by scientific literature [29, 34, 35]. Reducing co-morbidities may strengthen the battle against stunning, which would otherwise become a double or triple burden. It is also important to reinforce the need to incorporate intervention activities of the nutritional problem and diseases like diarrhea, respiratory infections, malaria, etc. [35].

This study shows, as the child's age increases, so are the likelihood that the child will be stunted. Scholarly articles in Ethiopia and elsewhere in other parts of the globe supported the finding [36–38]. As stunting has a constant and cyclical nature, inadequate dietary practice, weaning, lower and insufficient breast, and complementary feeding strategies have been weakened and become unsuccessful as the child's age increases, which further causes stunting. Another possible reason for the higher risk of stunning among older children could be the unhygienic preparation of additional food that exposes children to frequent infections. The area being studied is also exposed to the many kinds of infections and diarrheal diseases, which increase the risk of chronic malnutrition via reducing the access of these children to drinking water.

Besides, one interesting finding that emerged in this study was the number of livestock owned by the household became a strong correlate of stunting. The economic status measured indirectly by TLU implied that stunting can be addressed by considering the cultural and economic context of the area. Feeding the livestock products for the children could bring change in disease prevention. Though it was measured by different contexts in different studies, economic status was the major risk factor for stunting in Ethiopia [37, 39], and elsewhere in Africa [38]. It is well understood that poor people are suffering from poor diet, inadequate schooling, poor clothing, poor hygiene, and health, resulting in the children to suffer from failure in growth [37]. To tackle malnutrition, initiatives aiming to increase the number of animals in the nomadic community are therefore critical. It is also crucial to promote animal health by increasing awareness of animal disease prevention and control, by enhancing access to animal health facilities, and, most importantly, by piling up animal feed.

This research found that children born to households who obtained drinking water from unimproved water sources (rivers) were more likely their children to be stunted than their counterparts who obtained from improved water sources. Similar research in Ethiopia found

that households drinking water from an unprotected source were more stunted than their counterparts, corroborating this result [28, 37]. The lack of safe water causes multiple types of infection and diarrhoeal disease, which in turn raises chronic malnutrition. To tackle the problem of malnutrition in the area, improving access to better water sources is very necessary. Hence, this study bold out that to tackle stunting among children, non-nutrition-specific strategies have also paramount importance.

Stunting is found to be associated with the time of initiation of breastfeeding for the inborn child, in this study. Children who started breastfeeding immediately after birth within an hour whose mothers began breastfeeding suffered less from stunting. To improve the nutritional status of the infant, it is commonly recommended that children start breastfeeding immediately after birth. This may be because early breastfeeding leads to increased secretion and production of breast milk that will provide the baby with sufficient nutrients, such as colostrum [32]. Colostrum provides natural immunity to the infant and thereby decreases hypoglycemia and hypothermia, which in turn protects the infant's wellbeing [32, 40]. This study supports the WHO recommendation, which underlines the value of timely breastfeeding to children's health [33]. The results are backed by similar studies in Tigray, Northern Ethiopia [28]; in Indonesia [41]. These findings demonstrate the importance of early breastfeeding initiation as a means of early maternal care and the best food that can reduce the risk of stunting. Early breastfeeding is designated as one of the gateways to effective breastfeeding practice and ensures that infants obtain sufficient food [42]. Hence, health education should also be given to mothers on the benefits of early breastfeeding in improving the nutritional status of children. In providing a close follow-up on the matter, health extension workers and women's health development armies are vital [43].

Also, the current study showed that one of the important predictors of stunting was exclusive breastfeeding. Children who were not breastfed for at least 6 months exclusively were 2.57 times more likely to get stunted. A parallel can be drawn with scholarly articles [44–47]. The likely explanation is that for children whose digestive and immune systems are not yet mature, inappropriate timing for providing complimentary food will affect their nutritional status. The provision of food supplements may be a significant cause of malnutrition, particularly under unhygienic conditions [47]. To prevent infections that could hinder the development of the infant, exclusive breastfeeding is very necessary, particularly in the region where the sanitation status is very poor. Therefore, mothers should be advised to benefit from this and an enabling environment should be developed that promotes optimal breastfeeding.

This study indicated that one of the correlates of stunting was found to be a lack of adequate food diversity. It is 93% more likely that children who have eaten less optimal dietary diversity would be stunted. This finding is confirmed by several similar studies conducted in Ethiopia and elsewhere [48–50]. Therefore, this study demonstrates that malnutrition can be reduced by increasing the variety of complementary foods. Households should be educated and encouraged to provide appropriate and varied foods that can satisfy the need for energy and nutrients for the infant. Since a large number of cattle are owned by the pastoral group, supplying milk for their children is imperative.

Several projects have been implemented in Ethiopia to tackle malnutrition. One of the strategies was PSNP. However, the wealthier households are more likely to benefit from the PSNP than poorer households in Afar region [51]. The present finding indicated that PSNP didn't help in reducing stunting among children. The possible justification might be, the right target groups for such kinds of interventions, which are poor households were not sufficiently addressed in the program. Hence, to bring plausible effects on health and nutrition, addressing the target groups is very crucial.

The research has the limitations set out below. Due to the cross-section design of the research, we cannot declare a temporal association between stunting and other independent variables. Standard height/length measurement procedures have been used, but measurement errors, particularly among evaluators, are unavoidable. Besides, recall bias can occur in children who live in rural villages to report the age of the child. Nevertheless, if available, we have attempted to confirm the age stated in the immunization card.

## Conclusion

In conclusion, in the pastoralist community, the stunting situation was critically high, suggesting that stunting is still an issue of public health. The study found that stunting was linked to various nutrition-specific and non-nutrition-specific factors. Childhood age, household economic status, early initiation of breastfeeding, exclusive breastfeeding, and source of water supply. It is recommended to improve the economic status of households by preserving animal welfare and diversifying sources of income, by supporting optimal feeding practices for infants and young children, by complying with the WHO and national breastfeeding and complementary feeding guidelines. Protecting existing sources of water from potential pollution and increasing the coverage of safe sources of water in the region are also significant. Findings also suggest the importance of addressing income inequality when implementing nutrition strategies. According to the findings, comprehensive action on the underlying factors, such as economic status and access to improved water sources, is needed to achieve the SDG targets related to child nutrition. Otherwise, Ethiopia will falter to meet and an economic and health burden in the future generation will be inevitable. Generally, the findings of this study revealed that the etiologic factors of stunting are multifactorial. This means that implementing initiatives in a piecemeal fashion will significantly contribute to the persistence of malnutrition. A systematic and organized approach is thus needed for addressing the multiple and interconnected determinants of stunting throughout an individual's life cycle. The Countermeasures should be optimized according to evidence observed in the nomadic community, contextually with their way of life and socioeconomic status.

## Supporting information

**S1 Table. Afar version questionnaire.**
(DOCX)

**S2 Table. English version questionnaires.**
(DOCX)

**S1 Dataset.**
(SAV)

## Acknowledgments

The authors would like to extend their grateful acknowledgments to the University of Gondar, College of Medicine and Health Sciences, Institute of Public Health for providing ethical clearance. We also would like to appreciate the Afar Regional Health Bureau for facilitating the data collection process. Finally, we would like to extend our cordial appreciation to all the participants and data collectors.

## Author Contributions

**Conceptualization:** Mulugeta Gebreayohanes, Awrajaw Dessie.

**Data curation:** Awrajaw Dessie.

**Formal analysis:** Mulugeta Gebreayohanes, Awrajaw Dessie.

**Investigation:** Mulugeta Gebreayohanes, Awrajaw Dessie.

**Methodology:** Mulugeta Gebreayohanes, Awrajaw Dessie.

**Project administration:** Mulugeta Gebreayohanes, Awrajaw Dessie.

**Resources:** Mulugeta Gebreayohanes.

**Software:** Mulugeta Gebreayohanes, Awrajaw Dessie.

**Supervision:** Awrajaw Dessie.

**Visualization:** Awrajaw Dessie.

**Writing – original draft:** Awrajaw Dessie.

**Writing – review & editing:** Mulugeta Gebreayohanes, Awrajaw Dessie.

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
