## [Decision Letter · Decision Letter 0]

25 Nov 2020

PONE-D-20-23304

Prevalence of stunting and its associated factors among children 6-59 months of age in pastoralist community, Northeast Ethiopia: A community based cross sectional study

PLOS ONE

Dear Dr. Dessie,

Thank you for submitting your manuscript to PLOS ONE. After careful consideration, we feel that it has merit but does not fully meet PLOS ONE’s publication criteria as it currently stands. Therefore, we invite you to submit a revised version of the manuscript that addresses the points raised during the review process.

This article can be acceptable given that authors come-up with much better presentation of their findings and discussion. In particularly, the article needs a serious copy editing of the language. I suggest to follow the reviewers comments for revise and resubmit the manuscript.

We look forward to receiving your revised manuscript.

Kind regards,

Srinivas Goli, Ph.D.

Academic Editor

PLOS ONE

Journal Requirements:

2.We note that [Figure(s) 1] in your submission contain [map/satellite] images which may be copyrighted. All PLOS content is published under the Creative Commons Attribution License (CC BY 4.0), which means that the manuscript, images, and Supporting Information files will be freely available online, and any third party is permitted to access, download, copy, distribute, and use these materials in any way, even commercially, with proper attribution. For these reasons, we cannot publish previously copyrighted maps or satellite images created using proprietary data, such as Google software (Google Maps, Street View, and Earth). For more information, see our copyright guidelines: http://journals.plos.org/plosone/s/licenses-and-copyright.

1.    You may seek permission from the original copyright holder of Figure(s) [1] to publish the content specifically under the CC BY 4.0 license. 

3.In your Data Availability statement, you have not specified where the minimal data set underlying the results described in your manuscript can be found. PLOS defines a study's minimal data set as the underlying data used to reach the conclusions drawn in the manuscript and any additional data required to replicate the reported study findings in their entirety. All PLOS journals require that the minimal data set be made fully available. For more information about our data policy, please see http://journals.plos.org/plosone/s/data-availability.

4. Please provide a copy of your questionnaire in the original language and English and, as Supporting Information.

5.We suggest you thoroughly copyedit your manuscript for language usage, spelling, and grammar. If you do not know anyone who can help you do this, you may wish to consider employing a professional scientific editing service.  

6.We noticed you have some minor occurrence of overlapping text with the following previous publications, which needs to be addressed:

- https://bmcnutr.biomedcentral.com/articles/10.1186/s40795-019-0300-0

- https://journals.plos.org/plosone/article?id=10.1371%2Fjournal.pone.0195361

- https://mrmjournal.biomedcentral.com/track/pdf/10.1186/s40248-019-0188-1.pdf

In your revision ensure you cite all your sources (including your own works), and quote or rephrase any duplicated text outside the methods section. Further consideration is dependent on these concerns being addressed.

Additional Editor Comments (if provided):

This article can be acceptable given that authors come-up with much better presentation of their findings and discussion. In particularly, the article needs a serious copy editing of the language. I suggest to follow the reviewers comments for revise and resubmit the manuscript.

Reviewers' comments:

Reviewer's Responses to Questions

**Comments to the Author**

1. Is the manuscript technically sound, and do the data support the conclusions?

Reviewer #1: Yes

Reviewer #2: Yes

Reviewer #3: Yes

2. Has the statistical analysis been performed appropriately and rigorously? 

Reviewer #1: Yes

Reviewer #2: Yes

Reviewer #3: Yes

3. Have the authors made all data underlying the findings in their manuscript fully available?

Reviewer #1: Yes

Reviewer #2: Yes

Reviewer #3: Yes

4. Is the manuscript presented in an intelligible fashion and written in standard English?

Reviewer #1: Yes

Reviewer #2: Yes

Reviewer #3: No

5. Review Comments to the Author

Reviewer #1: The write-up needs to focus more on what new findings are coming up in the present study, and it needs to be presented better, in terms of discussion of its implications and how it affects the country as a whole.

Reviewer #2: This paper presents the Prevalence of stunting among children and its associated factors in Ethiopia based on primary data collected using a multi-stage sampling technique.

Overall, it is a good study and highlighted the critical issues of stunting among children and identified its associated factors in Ethiopia.

In the results section – while presenting the results, present only proportions, not the numbers of participants, avoid presenting both numbers as well as percentages. Readers can check it out the numbers if they want from the Tables. If it is presented in numbers, then, it is highly challenging for readers to understand the results.

In Table 2 information of morbidity among children has presented but not been discussed, the result shows one-fifth of children has two or more diseases, and it is important to discuss here, 20% is quite a large number.

Since the information of Diseases among children has been collected and has used in Table 2 as well, it is also advisable to use it in logistic regression and see whether the diseases will have any role to play in stunting among children.

Reviewer #3: The research work is technically sound and this paper can be accepted with some minor changes as discussed below.

1. The article needs copy editing with regard to language and ambiguity of points. For example, the sentence “The odds of stunting was increased, so does age of the child increased as compared to 6-11 months of children.” And at line 39. “………..livelihood strategies remain largely dependent on the climate system and susceptible to seasonal” at line 89 has an abrupt ending.

2. It was mentioned that total 555 households were interviewed (there were total 909 children from 554 households as can be seen from the analysis in table 1). How did you choose the children from these household what was the exclusion inclusion criteria? What was the method adopted to select the child? This need to be mentioned in the methodology part.

3. Definition of fully, partly and not vaccinated should be mentioned in footnotes. As per WHO “children aged 12–23 months, if the final primary vaccination is at 9 months of age – this is the most commonly chosen target population”. Therefore, it would be better to exclude children who have not completed 12 months from the analysis of immunization.

4. Under diseases in table 2 what are the childhood diseases covered in the study. A description of the diseases included in the study would be useful for the prospective reader.

5. Table no 3 there is row on decision making – “Decision making on use of money”. The answers are ‘mainly spouse” and “mainly husband”. If you interviewed the mother for information what would option “mainly spouse” mean, it would mean husband. These options need to be looked at.

6. In the discussion part at line 321 there is a sentence “Policies seeking to increase the number of animals in the nomadic culture are therefore crucial in combating malnutrition. Increasing animal health by raising awareness about the prevention and control of animal diseases, improving access to animal health facilities and, most importantly, piling up animal fodder is necessary.” Could not understand how that’s relevant here.

7. The referencing style is not uniform in the paper.

6. PLOS authors have the option to publish the peer review history of their article (what does this mean?). If published, this will include your full peer review and any attached files.

Reviewer #1: **Yes: **Ayantika Biswas

Reviewer #2: No

Reviewer #3: No

---

## [Author Response · Author response to Decision Letter 0]

14 May 2021

Author’s response for reviewers 

No Referee Comment Response Reference

1 Editor Please ensure that your manuscript meets PLOS ONE's style requirements, including those for file naming. The PLOS ONE style templates can be found at

 We have adhered to the style of PLOS ONE’s accordingly. It can be appreciated form the entire manuscript. 

2 Editor We note that [Figure(s) 1] in your submission contain [map/satellite] images which may be copyrighted. All PLOS content is published under the Creative Commons Attribution License (CC BY 4.0), which means that the manuscript, images, and Supporting Information files will be freely available online, and any third party is permitted to access, download, copy, distribute, and use these materials in any way, even commercially, with proper attribution. For these reasons, we cannot publish previously copyrighted maps or satellite images created using proprietary data, such as Google software (Google Maps, Street View, and Earth). For more information, see our copyright guidelines: http://journals.plos.org/plosone/s/licenses-and-copyright

Since we are unable to get permission from the copyright holder, we have removed the figure from the manuscript. We believed that description we provide in the manuscript is sufficient to define the study area. Page 6, Line 121-124. 

3 Editor In your Data Availability statement, you have not specified where the minimal data set underlying the results described in your manuscript can be found. We have uploaded the data set, according to the journal requirement. 

4 Editor Please provide a copy of your questionnaire in the original language and English and, as Supporting Information. We have provided accordingly the questionnaire. 

5 Editor We suggest you thoroughly copyedit your manuscript for language usage, spelling, and grammar. If you do not know anyone who can help you do this, you may wish to consider employing a professional scientific editing service. We have edited the language accordingly. It can be seem from the manuscript. 

6 Editor We noticed you have some minor occurrence of overlapping text with the following previous publications, which needs to be addressed:

- https://bmcnutr.biomedcentral.com/articles/10.1186/s40795-019-0300-0

- https://journals.plos.org/plosone/article?id=10.1371%2Fjournal.pone.0195361

- https://mrmjournal.biomedcentral.com/track/pdf/10.1186/s40248-019-0188-1.pdf

In your revision ensure you cite all your sources (including your own works), and quote or rephrase any duplicated text outside the methods section. Further consideration is dependent on these concerns being addressed.

 We have edited the manuscript to avoid the text overlapping with previous published works. We have made the necessary corrections by appropriate citation, quoting, and rephrasing the text. It can be seen from the manuscript. 

7 Reviewer #1 The authors need to drastically revamp the grammatical aspect of the article, especially in the ‘Introduction and background’.

 The language was extensively edited. You can see the whole manuscript.

 Reviewer #1 The authors need to specify what new findings are coming up in the study which are affecting stunting among the subjects, as compared to extant literature. So, the ‘discussion’ needs to focus more on what has already been found and how the current study adds on to the existing body of work, as opposed to just finding similarities with existing hypotheses.

 Thank you for the comment. We have tried to focus on the major findings and its implication with local context. The discussion section. 

 Reviewer #1 Concrete measures as to the alleviation of the existing situation and policy perspectives need to be focused on. The last two lines of the article seemed to touch upon the topic, but no sound policy recommendations were made on the basis of the study.

 Thank you for the comment. Based on the findings, we have provided plausible recommendations. Especially targeting underlying factors for stunting, following organized and collaborative approach in addressing stunting, and use of the local context, since the community is nomadic. Page 24: Line 401-417. 

 Reviewer #1 How the current situation affects the SDG attainment of the country as a whole, the relevant topic was mentioned in the introduction, but it was not followed up in the conclusion and discussion.

 Thank you for the comment, we have addressed the issue in the discussion and conclusion part of the manuscript accordingly. Discussion and conclusion part

 Reviewer #1 The analysis, specification of the data, methods and the tables are fine, but the background and the interpretation of the study need to be solidified. 

 Thank you for the comment. We have tried to strengthen the background and the result part, based on the comments given. 

9 Reviewer #2 In the results section – while presenting the results, present only proportions, not the numbers of participants, avoid presenting both numbers as well as percentages. Readers can check it out the numbers if they want from the Tables. If it is presented in numbers, then, it is highly challenging for readers to understand the results. Thank you for the comment, we have corrected accordingly. Result section

10 Reviewer #2 In Table 2 information of morbidity among children has presented but not been discussed, the result shows one-fifth of children has two or more diseases, and it is important to discuss here, 20% is quite a large number.

 Thank you for the comment, we have addressed discussed the morbidity result and its implication for nutritional intervention also indicated. Page 20, Line 326-320.

11 Reviewer #2 Since the information of Diseases among children has been collected and has used in Table 2 as well, it is also advisable to use it in logistic regression and see whether the diseases will have any role to play in stunting among children. We have checked the association of morbidity and stunting. However, it was not significant in the X2 test and we didn’t include in the logistic regression. 

12 Reviewer #3 1. The article needs copy editing with regard to language and ambiguity of points. For example, the sentence “The odds of stunting was increased, so does age of the child increased as compared to 6-11 months of children.” And at line 39. “………..livelihood strategies remain largely dependent on the climate system and susceptible to seasonal” at line 89 has an abrupt ending. Thank you for the comment, we have addressed the issue accordingly by undergoing through review of the manuscript. It can be seen from the entire manuscript. 

 The specific comments were also addressed, example Page 1, Line 70. 

13 Reviewer #3 2. It was mentioned that total 555 households were interviewed (there were total 909 children from 554 households as can be seen from the analysis in table 1). How did you choose the children from these household what was the exclusion inclusion criteria? What was the method adopted to select the child? This need to be mentioned in the methodology part. Thank you for the comment, If a household had two or more eligible children, the index child or the youngest one was included in the study. We have included this information in the manuscript accordingly. Page 7, Line 148-150.

14 Reviewer #3 3. Definition of fully, partly and not vaccinated should be mentioned in footnotes. As per WHO “children aged 12–23 months, if the final primary vaccination is at 9 months of age – this is the most commonly chosen target population”. Therefore, it would be better to exclude children who have not completed 12 months from the analysis of immunization. Thank you for the comment. We have provided the definition in the method section. Since, we define the status of vaccination by completing all doses of vaccine for the given age as fully vaccinated. Missing at least one dose of vaccine for the given age as partially vaccinated and not taking any doses of vaccine for the given age as not vaccinated. We believe excluding the children who have not completed 12 months is not important. Page 8, Line 182-187. 

15 Reviewer #3 4. Under diseases in table 2 what are the childhood diseases covered in the study. A description of the diseases included in the study would be useful for the prospective reader. We have included the description of the diseases found in the study, accordingly. Page 12, Line 238-241. 

16 Reviewer #3 5. Table no 3 there is row on decision making – “Decision making on use of money”. The answers are ‘mainly spouse” and “mainly husband”. If you interviewed the mother for information what would option “mainly spouse” mean, it would mean husband. These options need to be looked at. Thank you for the concern, spouse in this study was to mean wife. To avoid the confusion we have use the word wife instead. Table 3.

17 Reviewer #3 6. In the discussion part at line 321 there is a sentence “Policies seeking to increase the number of animals in the nomadic culture are therefore crucial in combating malnutrition. Increasing animal health by raising awareness about the prevention and control of animal diseases, improving access to animal health facilities and, most importantly, piling up animal fodder is necessary.” Could not understand how that’s relevant here. Thank you for the comment. The reason why we included the statement was due to the fact that economic status measured by Tropical Live Stock (TLU) was found to be one of the correlates of stunting. Households who were poor in economic status were at higher risk of stunting. Since the study was conducted among nomadic community who are dependent on animas as livelihood, we have stressed the significance of awareness creation to improve animal health and providing sufficient fodder to increase the size of animal population. 

18 Reviewer #3 7. The referencing style is not uniform in the paper. We have used the reference style recommended by the journal and we have amended the irregularity in the style.

---

## [Decision Letter · Decision Letter 1]

8 Jul 2021

PONE-D-20-23304R1

Prevalence of stunting and its associated factors among children 6-59 months of age in pastoralist community, Northeast Ethiopia: A community based cross sectional study

PLOS ONE

Dear Dr. Dessie,

Thank you for submitting your manuscript to PLOS ONE. After careful consideration, we feel that it has merit but does not fully meet PLOS ONE’s publication criteria as it currently stands. Therefore, we invite you to submit a revised version of the manuscript that addresses the points raised during the review process.

ACADEMIC EDITOR: This paper can be considered after a minor revision. Please address the review 3 comments. Hoping to see the revised version. 

We look forward to receiving your revised manuscript.

Kind regards,

Srinivas Goli, Ph.D.

Academic Editor

PLOS ONE

Journal Requirements:

Additional Editor Comments (if provided):

This paper can be considered after a minor revision. Please address the review 3 comments. Hoping to see the revised version.

Reviewers' comments:

Reviewer's Responses to Questions

**Comments to the Author**

1. If the authors have adequately addressed your comments raised in a previous round of review and you feel that this manuscript is now acceptable for publication, you may indicate that here to bypass the “Comments to the Author” section, enter your conflict of interest statement in the “Confidential to Editor” section, and submit your "Accept" recommendation.

Reviewer #2: All comments have been addressed

Reviewer #3: (No Response)

2. Is the manuscript technically sound, and do the data support the conclusions?

Reviewer #2: Yes

Reviewer #3: Yes

3. Has the statistical analysis been performed appropriately and rigorously? 

Reviewer #2: Yes

Reviewer #3: Yes

4. Have the authors made all data underlying the findings in their manuscript fully available?

Reviewer #2: Yes

Reviewer #3: (No Response)

5. Is the manuscript presented in an intelligible fashion and written in standard English?

Reviewer #2: Yes

Reviewer #3: No

6. Review Comments to the Author

Reviewer #2: The author has addressed all the questions raised in the previous rounds of review; the paper may be accepted with minor correction.

Reviewer #3: The manuscript requires copy editing of the language and better presentation. The authors may seek professional help if needed.

If Line 95 "The global nations' sustainable development goals (SDGs) " it should be "the United Nation's SDGs....."

Table no 1 :"Educational status of a mother" it should be "Educational status of mother"

Line 244 &b 245 "80.5% of the majority of children were breastfed exclusively for at least six months" It should be "majority (80.5%) of the children were breastfed exclusively for at least six months". In many places use of parenthesis is missing in the manuscript.

Please cross check the figures in the table and the in the description. In a few places the figures in description are mismatching with what is in the table. For example; in line 240 "This study showed that 56.5% of the children were " but the table 1 says the percentage is 56.3 and in Line no 260 "households concerning the use of money (71.2%)" Table 3 shows this to be 71.3 %. Such mistakes should be avoided.

Under morbidity, what are the diseases covered in the study are still not mentioned anywhere in the paper.

7. PLOS authors have the option to publish the peer review history of their article (what does this mean?). If published, this will include your full peer review and any attached files.

Reviewer #2: No

Reviewer #3: No

---

## [Author Response · Author response to Decision Letter 1]

4 Aug 2021

Response to reviewers 

No Referee Comment Response Reference

1 Editor Please review your reference list to ensure that it is complete and correct. If you have cited papers that have been retracted, please include the rationale for doing so in the manuscript text, or remove these references and replace them with relevant current references. Any changes to the reference list should be mentioned in the rebuttal letter that accompanies your revised manuscript. If you need to cite a retracted article, indicate the article’s retracted status in the References list and also include a citation and full reference for the retraction notice. We have reviewed the reference list and it is correct. Retracted papers not cited in this manuscript. 

We have changed reference number 22 in page 8. The former reference in the old version of the manuscript was unpublished work. We have replaced it with published articles. 

The reference number 30 and 31 in old version were the same reference, but appears as different references in the manuscript. This issue was addressed accordingly. 

2 Reviewer #3 The manuscript requires copy editing of the language and better presentation. The authors may seek professional help if needed.

 Thank you very much for the comment. Copy editing of the language has been extensively done accordingly. It can be seen from the entire manuscript. 

3 Reviewer #3 If Line 95 "The global nations' sustainable development goals (SDGs) " it should be "the United Nation's SDGs....."

 Thank you. We have corrected it accordingly. Page 1; Line 95.

4 Reviewer #3 Table no 1 :"Educational status of a mother" it should be "Educational status of mother" We have corrected it accordingly. Table 1. 

5 Reviewer #3 Line 244 &b 245 "80.5% of the majority of children were breastfed exclusively for at least six months" It should be "majority (80.5%) of the children were breastfed exclusively for at least six months". In many places use of parenthesis is missing in the manuscript. Thank you very much for the comment. It has been modified accordingly. Throughout the manuscript such kind of corrections have been made. Page 13; Line 247. 

6 Reviewer #3 Please cross check the figures in the table and the in the description. In a few places the figures in description are mismatching with what is in the table. For example; in line 240 "This study showed that 56.5% of the children were " but the table 1 says the percentage is 56.3 and in Line no 260 "households concerning the use of money (71.2%)" Table 3 shows this to be 71.3 %. Such mistakes should be avoided. Thank you very much. We have made the corrections accordingly. Now the figures in the text description and in the respective tables are speaking the same. 

7 Reviewer #3 Under morbidity, what are the diseases covered in the study are still not mentioned anywhere in the paper. Thank you for the comment. We have addressed the issue accordingly. Page 13; Line 253-255. 

8 Reviewer #3 On page 8, In the definition of fully vaccinated- children who had received a vaccination against tuberculosis (BCG), three doses each of the DPT and polio vaccines, and a measles vaccination by 12 months of age…… In standard practice, it considers the children aged 12 to 23 months for full immunisation; even WHO also suggests the period 12-23 months to calculate full immunisation….it is better to follow the standard practice so the results can also be comparable with other studies as well. 

 Thank you very much for the comment. We have modified the definition accordingly. The status of vaccination has been computed according to the WHO standard, by considering the age group from 12-23 months. Page 8; Line 182-189. Table 2.

---

## [Editor Report · Decision Letter 2]

16 Aug 2021

Prevalence of stunting and its associated factors among children 6-59 months of age in pastoralist community, Northeast Ethiopia: A community based cross sectional study

PONE-D-20-23304R2

Dear Dr. Dessie,

We’re pleased to inform you that your manuscript has been judged scientifically suitable for publication and will be formally accepted for publication once it meets all outstanding technical requirements.

Kind regards,

Srinivas Goli, Ph.D.

Academic Editor

PLOS ONE

Additional Editor Comments (optional):

Considering my own reading and the reviewers opinion, I am recommending this paper for publication in PLOS subject to minor revision from authors.
---

## [Editor Report · Acceptance letter]

24 Jan 2022

PONE-D-20-23304R2 

Prevalence of stunting and its associated factors among children 6-59 months of age in pastoralist community, Northeast Ethiopia: A community-based cross-sectional study 

Dear Dr. Dessie:

I'm pleased to inform you that your manuscript has been deemed suitable for publication in PLOS ONE. Congratulations! Your manuscript is now with our production department. 

Kind regards, 

on behalf of

Dr. Srinivas Goli 

Academic Editor

PLOS ONE